



# Enhancing LSTM-based streamflow prediction with a spatially distributed approach

Qiutong Yu[1], Bryan A. Tolson[1], Hongren Shen[1], Ming Han[2], Juliane Mai[3], Jimmy Lin[4]

[1]Department of Civil and Environmental Engineering, University of Waterloo, Waterloo, ON, Canada
[2]Water Resources, Ontario Power Generation Inc., Niagara Falls, ON, Canada
[3]Department of Earth and Environmental Science, University of Waterloo, Waterloo, ON, Canada
[4]David R. Cheriton School of Computer Science, University of Waterloo, Waterloo, ON, Canada

*Correspondence to*: Qiutong Yu (q45yu@uwaterloo.ca)

**Abstract**

Deep learning (DL) algorithms have previously demonstrated their effectiveness in streamflow prediction. However, in hydrological time series modelling, the performance of existing DL methods is often bound by limited spatial information, as these data-driven models are typically trained with lumped (spatially-aggregated) input data. In this study, we propose a hybrid approach, namely the Spatially Recursive (SR) model, that integrates a lumped long short-term memory (LSTM) network with a physics-based hydrological routing simulation for enhanced streamflow prediction. The lumped LSTM was

trained on the basin-averaged meteorological and hydrological variables derived from 141 gauged basins located in the Great Lakes region of North America. The SR model involves applying the trained LSTM at the subbasin scale for local streamflow predictions which are then translated to the basin outlet by the hydrological routing model. We evaluated the efficacy of the SR model on predicting streamflow at 224 gauged stations across the Great Lakes region and compared its performance to that of the standalone lumped LSTM model. The results indicate that the SR model achieved performance

levels on par with the lumped LSTM in basins used for training the LSTM. Additionally, the SR model was able to predict streamflow more accurately on large basins (e.g., drainage area greater than 1000 km$^2$), underscoring the substantial information loss associated with basin-wise feature aggregation. Furthermore, the SR model outperformed the lumped LSTM when applied to basins that were not part of the LSTM training (i.e., pseudo-ungauged basins). The implication of this study is that the lumped LSTM predictions, especially in large basins and ungauged basins, can be reliably improved by

considering spatial heterogeneity at finer resolution via the SR model.

## 1 Introduction

Reliable streamflow prediction is critical in water resources management. Following recent developments in Artificial Intelligence (AI), an increasing number of hydrological studies have focused on adopting deep learning (DL) techniques, such as long short-term memory (LSTM), to improve basin-scale streamflow prediction compared to traditional physically-

based hydrologic models and conventional machine-learning (ML) algorithms (Kratzert et al., 2018; Frame et al., 2021;



Gauch et al., 2021). LSTM is a type of Recurrent Neural Network (RNN) that can capture long-term dependencies in hydrological time-series data and has demonstrated promising results in tasks such as streamflow prediction (Kratzert et al., 2018; Gauch et al., 2021), precipitation forecasting (Tao et al., 2021), and drought monitoring (Wu et al., 2022). A recent large-sample model intercomparison study, namely the Great Lakes Runoff Intercomparison Project in the Great Lakes

region (GRIP-GL; Mai et al., 2022), showed that LSTM model exhibited significant superiority in streamflow predictions compared to 12 other physically-based hydrological models, regardless of whether they were lumped or spatially distributed (Mai et al., 2022). The development of LSTM in hydrology has been driven by the need for more accurate and sophisticated models that can handle the complex and non-linear relationships in hydrological processes.

Despite of the recent popularity of data-driven modelling (e.g., ML and DL models) in hydrological modelling studies,
process-based hydrological models (physically-based or conceptual model) continue to be used for operational streamflow forecasting. In contrast with data-driven prediction, traditional process-based models often rely more on spatially distributed representation of the region or basin being simulated. They utilize gridded meteorological forcings and, more importantly, break up the basin into various smaller response units such as grid cells (fully-distributed model) or subbasins (semi-distributed model). Compared to lumped models, distributed models take into account spatial variability at a finer resolution
and also incorporate the routing process within the simulated basin. With the recognition that data-driven models and process-based models possess distinct advantages, the hybridisation of these two types of models has drawn growing attention in environmental modelling studies. Hybrid models can be categorized into two primary structural types: serial and parallel. In most cases, serial hybrid model involves the sequential coupling of one data-driven model and one process-based model (Hunt et al., 2022). This is typically achieved by feeding (to train) a data-driven model with the outputs of a process-
based model (Frame et al., 2021; Liu et al., 2021; Nevo et al., 2022; Zhang et al., 2022), which implies that the data-driven model usually serves as a post-processor within a hybrid modelling workflow. In a parallel hybrid model, the data-driven model and process-based model are integrated in parallel with each simulating different processes (Slater et al., 2023). In general, these hybrid modelling approaches allow researchers, to a certain extent, to incorporate spatial variability of input variables into the data-driven prediction scheme.

It is widely acknowledged that having ample training data is advantageous for DL models. Kratzert et al. (2018) argued that training a local LSTM streamflow model at an individual gauged basin is an inferior approach compared to a training a regional LSTM model over many gauged basins. They trained a single LSTM model for lumped rainfall-runoff simulation, on a large sample of 241 basins using meteorological forcing data and static basin attributes and then compared the performance of this regionally-trained LSTM streamflow model to that of individual local LSTM streamflow models trained
separately for each of the 241 basins. The results revealed that the regionally-trained LSTM model was able to outperform the local LSTM models. Nevertheless, in previous studies regarding LSTM-based streamflow prediction (to the best of our knowledge), most regionally-trained LSTM models only consider the spatial heterogeneity between training basins where LSTM inputs are at the lumped training basin scale. That is, each attribute (i.e., LSTM input variable) is computed for the entire basin and the basin is considered a single response unit such that streamflow is only predicted at the outlet of the basin



(Kratzert et al., 2018, 2019; Feng et al., 2020; Gauch et al., 2021; Xie et al., 2022; Arsenault et al., 2023; Tang et al., 2023; Pokharel et al., 2023). Typically, this involves cropping the gridded dynamic input variables (e.g., precipitation and temperature) to the basin polygon and calculating a basin-average (or weighted average) to produce the lumped time series of the dynamic input variables (Lees et al., 2022). The rationale behind employing a lumped model is due to the architectural limitation of LSTM networks which are not compatible with gridded data (i.e., image-like data) with various shapes (i.e.,

basin outlines), as inputs. Additionally, lumped model enables effective learning of static basin attributes, such as drainage area and frequency of high precipitation (Kratzert et al., 2019). However, the aggregation of climatic forcings results in neglecting the heterogenous spatial distribution of various rainfall events. A study by Hunt et al. (2022) explains that a lumped LSTM probably failed to appropriately characterize rainfall over a large and arid basin due to averaging rainfall to the basin scale. Wang and Karimi (2022) argue that lumped LSTM rainfall-runoff models are unable to fully utilize the

spatial variability of input features. In their experiments, the spatial variability of rainfall was represented by a 20-element vector feature. For each of the 10 basins where they trained the LSTM, the vector consists of rainfall data at all hydrological response units within the basin. Their results show that LSTM trained on spatial-distributed rainfall data outperformed those driven by basin-averaged rainfall data. Nonetheless, their method does not clarify how to generalize the process of supplying the LSTM with spatially distributed rainfall information in the context of predicting outcomes in ungauged basins (PUB).

This study aims to identify a viable approach for effectively implementing LSTM-based streamflow prediction with spatially-distributed inputs, which can be easily generalized and thus applied to out-of-sample tests (i.e., untrained and ungauged basins). In pursuit of this goal, we propose the Spatially Recursive (SR) model. The SR model first employs a lumped LSTM (trained on a large sample of basins) to predict local streamflow at subbasins discretized from the basin of interest. Then, it utilizes a semi-distributed hydrologic routing-only sub-model to route subbasin streamflow to the basin

outlet. The LSTM is considered spatially recursive because it is trained at the basin scale and further applied at the subbasin scale to incorporate finer-resolution forcing data and subbasin attributes. Distinct to previous studies using serial hybrid models, the hydrologic routing-only sub-model in the SR model is utilized as the post-processor for the LSTM-predicted outputs at subbasin scale. In this study, we use the lumped LSTM predictions as the benchmark to compare with the SR model results.

The paper is structured as follows: Section 2 provides a description of the SR model, datasets, and experimental design. Section 3 presents key results and discussion. Finally, Section 4 concludes the findings and outlines future work.

## 2 Material and Methods

### 2.1 Overview of Spatially Recursive (SR) model

The proposed SR model is composed of three workflow components (see Fig.1), a regional LSTM for basin outlet

streamflow prediction that is trained using a large sample of basins, a vector-based lake-river routing network that breaks up basins into subbasins, and a process-based routing sub-model that only simulates the movement of LSTM predicted





subbasin-level streamflow through the routing network. The fundamental concept of the SR model is to firstly employ a regional LSTM to predict local streamflow at each subbasin outlet (delineated in the lake-river routing network) for a basin where streamflow at the basin outlet is of interest. Note that local streamflow for a subbasin is defined as the streamflow at

the subbasin outlet that would occur if there was no upstream subbasins. Then, the predicted streamflow at each subbasin outlet serves as input to the routing-only sub-model, which simulates how water is transported through the lake-river routing network and ultimately the streamflow at the basin outlet.

In general, none of the three workflow components are novel when considered individually, as numerous existing studies have explored and assembled them in various ways. Example of vector-based watershed discretization include the MERIT-

hydro global database (Yamazaki et al., 2019) and the North American Lake-River Routing Product (NALRP; Han et al., 2020). Such routing networks should have typically featured a finer spatial resolution in order to break up each basin polygon into subbasins, at least for majority of the streamflow gauging stations used for LSTM training. Furthermore, example of a hydrologic routing model includes the routing models in Han et al. (2020) and Mizukami et al. (2016). The distinctive aspect of our research lies in the combination of components to form the SR model. The following subsections

explain the specifics of the components we selected in this study to demonstrate the spatially recursive modelling approach.

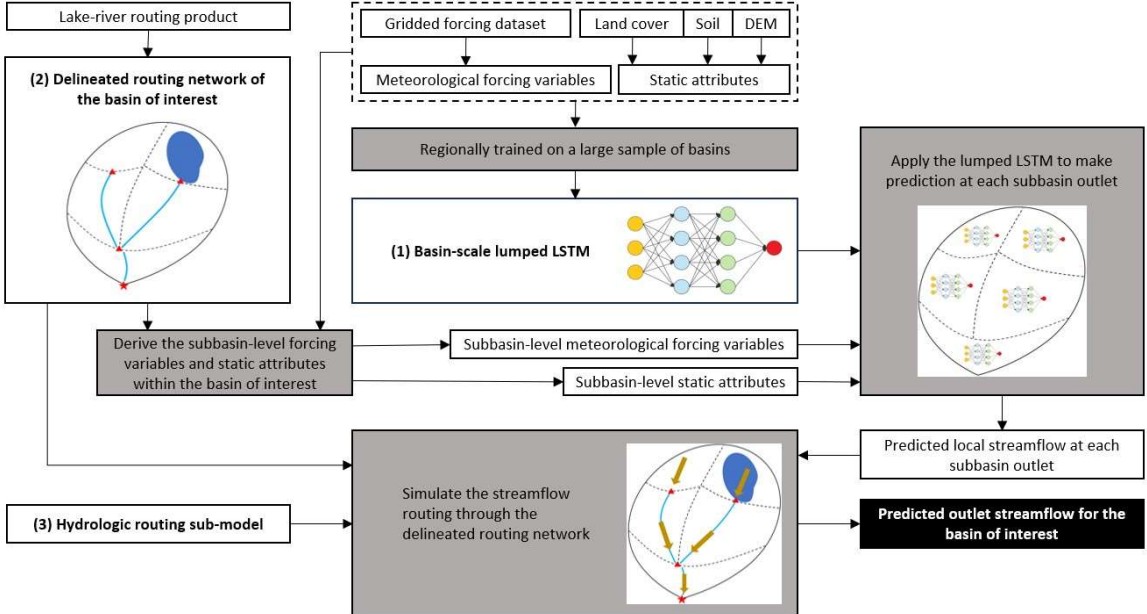

**Figure 1.** The spatially recursive model workflow for an arbitrary basin of interest. Note the workflow is model-agnostic, where the predictor model (lumped LSTM) can be replaced by another data-driven model. The three components of the SR model are numbered in
the order that they are applied in the workflow.





### 2.1.1 GRIP-GL Lumped LSTM

In this study, we replicated the lumped LSTM built for streamflow prediction in the Great Lakes region of North America by Mai et al. (2022). That is, we trained our version of the lumped LSTM model using the same hyperparameters and input features as the LSTM model trained in the GRIP-GL project. Only a brief summary of the model setup, training, and testing procedures of the GRIP-GL LSTM is presented here, and we refer the reader to the supplement paper of Mai et al. (2022), particularly Section S3.1, for the details of hyperparameter tuning and the full list of all model input features.

Following the training procedures outlined by Mai et al. (2022), our trained model is an ensemble of 10 LSTM models with the same architecture but different random seeds, and the final prediction is the average of the 10 models' outputs. Each LSTM model was simultaneously trained on 141 gauged basins (also referred to as 'calibration basins' in the GRIP-GL project) located in the Great Lakes region, over the period spanned from January 2000 to December 2010 (referred to as 'calibration period' in GRIP-GL). The LSTM model was constructed using the NeuralHydrology Python library (Kratzert et al., 2022). It was implemented to conduct sequence-to-one prediction, that is, the LSTM model predicts the average streamflow for a single day based on the input sequence of the previous 365 days of data.

The input features were derived for each basin, which include the target variable (observational streamflow at the daily time scale), 9 dynamic variables (meteorologic forcings), and 30 static basin attributes describing soils, topography, land cover types, and climate. The observed discharge data is from either Water Survey Canada (WSC) or United States Geological Survey (USGS). The meteorologic forcings and climatic attributes were taken from the Regional Deterministic Reanalysis System Version 2 (RDRS-v2) (Gasset et al., 2021). RDRS-v2 is a gridded reanalysis product which covers North America with a 10 km by 10 km spatial resolution on an hourly time step and this product was downloaded for the region from the CaSPAr archive (Mai et al., 2020). Soil attributes were derived from the Global Soil Dataset for Earth System Models (GSDE) (Shangguan et al., 2014). Topological attributes (e.g., mean elevation, mean slope) were computed from the HydroSHEDS digital elevation model (DEM) product (Lehner et al., 2008). And the North American Land Change Monitoring System (NALCMS) product was used to derive the land cover attributes. The target variable and dynamic variables were aggregated from hourly to daily timescale. All dynamic variables and static attributes were spatially averaged for each gauged basin in the study. All input features derived for our replication of the GRIP-GL LSTM model were completely consistent with the derived input features in the original GRIP-GL study. This consistency check against GRIP-GL was important because our LSTM input derivation scripts are reapplied here for numerous and generic smaller subbasin polygons. Most importantly, the resultant trained GRIP-GL LSTM model rebuilt here generated practically identical quality hydrographs as the GRIP-GL LSTM in Mai et al. (2022) with differences in the median KGE performance metric of less than 0.01 (due only to different random seeds used in training).

After the model training, the lumped LSTM model was evaluated in three validation experiments according to the testing procedures outlined by Mai et al. (2022). These three validation experiments are used consistently throughout this study and are listed as follows:



1. *Temporal validation*, conducted for 141 calibration basins which were used to train the LSTM, predicting the daily streamflow over the period from January 2011 to December 2017 (referred as 'validation period' in GRIP-GL).

2. *Spatial validation*, conducted for 71 basins which were not used for model training (referred as 'validation basins' in GRIP-GL), predicting the daily streamflow over the training/calibration period (January 2000 to December 2010).

3. *Spatiotemporal validation*, conducted for the 71 validation basins, predicting the daily streamflow over the validation period (January 2011 to December 2017).

Note that in this study we used the term 'validation' and 'testing' interchangeably in order to be consistent with the experimental design of GRIP-GL, and it is not same as the 'validation' terminology normally used in machine learning applications.

### 2.1.2 Spatially distributed prediction based on a lake-river routing network

The lake-river routing network for an arbitrary basin defines the connectivity between the lakes/reservoirs, river channels and subbasins, as well as initial values for subbasin, lake, and channel characteristics required for running a spatially distributed hydrological simulation. A routing product is defined as a collection of routing networks covering large geographic regions and all included networks in a routing product should be delineated using the same source Geographic Information System (GIS) products (e.g., Lake polygons, DEM etc.) (Han et al., 2023).

In this study, we tested our SR model with two routing products, the GRIP-GL common routing product (used in Mai et al. (2022)), and the North American Lake-River Routing Product v2.1 (NALRP; Han et al., 2020). Both products were generated by the BasinMaker Python library (Han et al., 2023), which supports delineation of vector-based routing networks from any DEM and user-defined lake polygons. The GRIP-GL common routing product was derived from the HydroSHEDS DEM, with a spatial resolution of 3 arcseconds. The river network and subbasin (discretization) were defined by a constant flow accumulation threshold of 5000. That is, for a given point of interest, the contributing drainage area would be at least 5000 DEM cells, which corresponds to approximately 40.5 km$^2$. On the other hand, the NALRP was produced based on the MERIT DEM (Yamazaki et al., 2019) with the same 3 arcseconds resolution, and the value of flow accumulation threshold is 2000 DEM cells (approximately 16.2 km$^2$). Additionally, attributes of lakes were taken from the HydroLAKES database (Messager et al., 2016) for both the GRIP-GL routing product and the NALRP. While the NALRP product included all lakes in HydroLAKES, the GRIP-GL routing product does not include small lakes with an area less than 5 km$^2$.

We use the GRIP-GL routing product directly in order to replicate in our SR model the precise routing networks Mai et al. (2022) used for their semi-distributed hydrological models. In contrast, we used the NALRP product here to provide flexibility to run our SR model in non-GRIP-GL basins and to evaluate SR-model sensitivity to alternative routing model configuration decisions. The BasinMaker library includes post-processing functions to further simplify the routing networks in the NALRP. When applying simplification to the routing network, the resolution of the routing network was reduced (Han et al., 2023) by incorporating fewer vector elements (river channels, subbasin polygons etc.) into the routing network. For





example, users can specify a minimum lake area threshold to remove the lakes with an area smaller than the threshold, and a minimum subbasin drainage area (MDA) threshold to merge subbasins. The MDA parameter provides the option to control the degree of simplification. It should be noted that, not all subbasins with drainage area smaller than the MDA threshold

will be merged. Figure 2 shows a single basin discretized into three example lake-river routing networks, including the GRIP-GL routing network in Figure 2A), the original high-resolution NALRP network in Figure 2B, and a simplified NALRP network in Figure (derived from the original NALRP network by applying BasinMaker functions). Note that our lake subbasins include only one lake which is completely contained within the subbasin boundary and our non-lake subbasins only have a single channel reach.


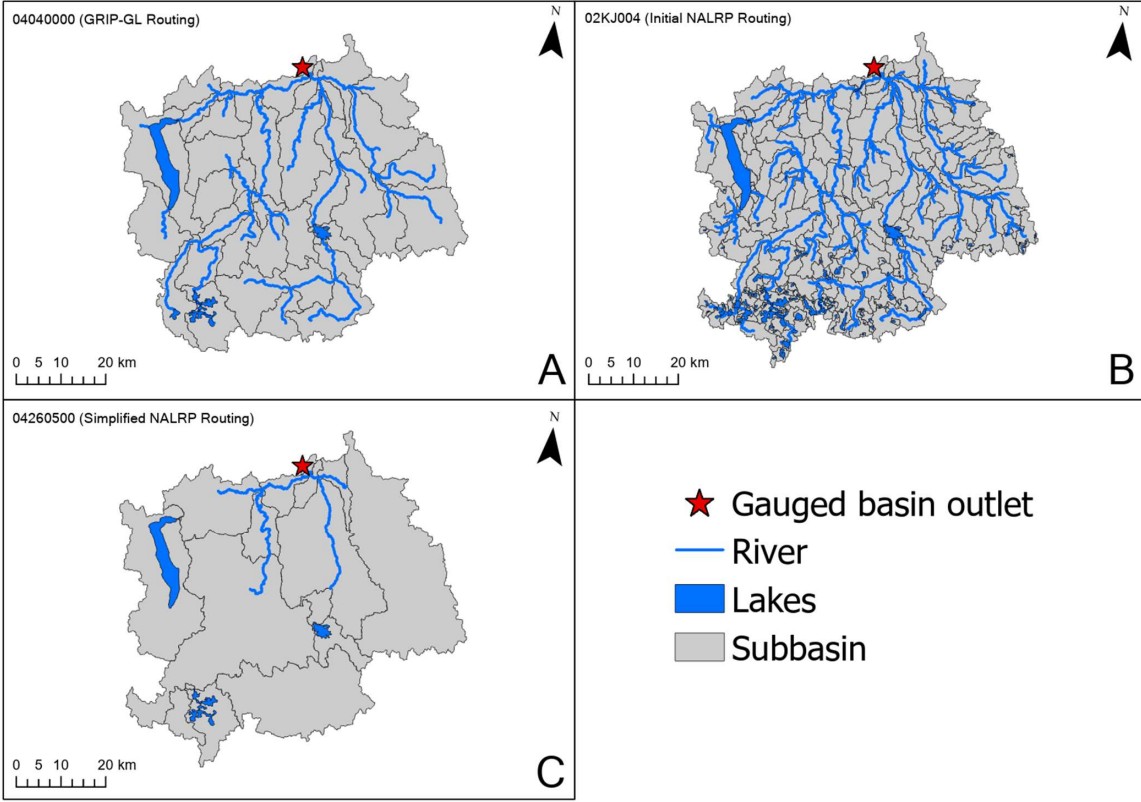

**Figure 2.** An example lake-river routing network for the Ontonagon River Watershed (USGS gauge ID: 04040000), which is one of the GRIP-GL calibration basins. (A) The network from the default GRIP-GL common routing product. (B) The network from the initial NALRP without simplification. (C) The simplified NALRP network delineated by a minimum lake area of 5 km$^2$ and a minimum subbasin

drainage area of 500 km$^2$.



As shown in Fig. 1, given the trained LSTM and the corresponding lake-river routing network discretizing a basin into subbasins, LSTM input features at the subbasin level need to be derived. This derivation is based on the original geospatial data and spatiotemporal data (see Section 2.1.1) and thus leverages the inherent spatial variability in the original data (as opposed to simply applying the basin-scale average input features in a basin to all the subbasins in that basin). With
subbasin-level LSTM input features, the lumped LSTM is then deployed in each subbasin to predict the local subbasin outlet streamflow.

### 2.1.3 Routing-only mode in the Raven hydrological modelling framework

In this study, we constructed a physically-based routing-only sub-model in the Raven hydrological modelling framework (Craig et al., 2020) to move local subbasin outlet streamflow through the routing network in each basin. Note the lake-river
routing network generated by BasinMaker incorporates all the necessary Raven routing model inputs and parameters (e.g., channel roughness, lake outlet characterization).

The intermediate results from our previous step in Section 2.1.2 (i.e., the LSTM local streamflow predictions at subbasin-level) can be seen as distributed subbasin-specific streamflow fluxes at various points within a routing network of streams and lakes (if any). The routing-only mode in the Raven framework can simulate the routing of distributed surface runoff
(Han et al., 2020; Craig et al., 2020) but here we use it for the first time to route local subbasin outlet streamflow. This is accomplished by representing the local streamflow input (at each subbasin) as hourly precipitation instantly flushing to the subbasin outlet. The routing model combines this local streamflow at the hourly timestep with the upstream subbasin streamflow that were routed from the subbasin inlet to the subbasin outlet. This process continues from upstream to downstream subbasins to the basin outlet. The routing model time step is hourly even though input streamflow from the
LSTM are daily averages. The simulated hourly streamflow at each basin outlet (stream gauge) is aggregated to daily streamflow which is the final prediction of the SR model.

The routing model is initialized with the lakes filled to the crest outlet elevation (point of zero outflow). For lake subbasins, the flushing operation sends all local subbasin streamflow into the lake instantly (rather than the subbasin outlet at the outlet of the lake) and then it becomes subject to the lake routing process in Raven. Since, water area is an input attribute in our
LSTM, the LSTM has implicitly been trained to at least partially reflect lake routing impacts. Hence, our approach within Raven means that for lake subbasins, the local subbasin streamflow delivery to the subbasin outlet is only approximate as this typically small fraction of the total streamflow reaching the subbasin outlet has lake routing impacts applied in two ways instead of only once. This approximation is unavoidable within Raven modelling system but to be clear, the impacts are negligible given that for most lakes, local subbasin streamflow is only a small fraction of the total streamflow entering the
lake considering all upstream subbasins.

The Raven framework provides the options to manipulate the routing algorithms and to calibrate routing-related parameters. In this study, we utilized the default configurations of the Raven framework and refrained from calibrating the routing-only sub-model. We selected the diffusive wave channel routing option (where an analytical solution to the diffusive wave





equation is used to relate inflow and outflow in each reach) and level-pool outflows from lakes are assumed to be governed
by the broad-crested weir equation. Since subbasin-level streamflow is predicted by a calibrated (trained) lumped LSTM
model, we assume that routing calibrated fluxes through a reasonably configured default routing model will typically yield
reasonable quality results. Effectively, this approach provides a lower-bound estimate of the SR model performance given
the routing model parameters are uncalibrated.

**2.2 Selection of additional gauging locations for concept validation**

We posit that the effective scale of LSTM prediction (i.e., generalizability on various watershed sizes) might be affected by
the range and distribution of the drainage area of the training/calibration basins, due to the variation of streamflow pattern in
watersheds with different size. For instance, hydrological responses in small watersheds tend to be raging and flashy
(Camera et al., 2020). The lumped LSTM was trained exclusively on basins with a drainage area exceeding 200 km$^2$ in
accordance with the selection criteria of gauging basins in the GRIP-GL project. Considering the watershed delineation
schemes (GRIP-GL routing and simplified NALRP) deployed in this study, many of the delineated subbasins would have a
drainage area smaller than 100 km$^2$. Furthermore, the GRIP-GL basins show a skewed distribution in terms of sizes, with
109 out of the 141 calibration basins having a drainage area between 200 to 2000 km$^2$. Similarly, 56 out of the 71 validation
basins fall within this range. To assess the LSTM performance at subbasin-level (smaller than 200 km$^2$) and larger
watersheds (greater than 2000 km$^2$), we selected 12 additional gauging basins (not used in the GRIP-GL) in the Great Lakes
region (within the bounding box defined by the minimum and maximum latitudes and longitudes of the GRIP-GL drainage
basin) for spatial and spatiotemporal validation. These basins (summarized in Table 1) include 4 small basins with a drainage
area below 100 km$^2$ and 8 large basins ranging in size from 2000 to 7000 km$^2$.

**Table 1**. Summary of the 12 Non-GRIP-GL gauging basins selected for this study.

| WSC or USGS station number | Description | Drainage Area (km$^2$) |
|---|---|---|
| 02HC017 | Etobicoke creek at Brampton, ON, Canada | 69 |
| 01415000 | Tremper kill near Andes, NY, USA | 86 |
| 04LA006 | Mollie River at Highway No.144, ON, Canada | 93 |
| 04105700 | Augusta Creek near Augusta, MI, USA | 95 |
| 05129115 | Vermilion River near Crane Lake, MN, USA | 2343 |
| 02KF001 | Mississippi River at Fergusons Falls, ON, Canada | 2660 |
| 02KJ004 | Dumoine (Riviere) A La Sortie Du Lac Robinson, QC, Canada | 3760 |
| 02KB001 | Petawawa River near Petawawa, ON, Canada | 4120 |
| 04260500 | Black River at Watertown, NY, USA | 4827 |
| 01529950 | Chemung River at Corning, NY, USA | 5195 |
| 04LA002 | Mattagami River near Timmins, ON, Canada | 5570 |
| 04LF001 | Kapuskasing River at Kapuskasing, ON, Canada | 6760 |






All Non-GRIP-GL gauged basins are selected based on the following additional criteria:

1.  The basin is not heavily regulated by dams or reservoirs.
2.  The basin has less than 5% of missing data in streamflow observation for the study period.
3.  The gauge ID at the basin outlet is included in the NALRP and thus defines a pre-existing routing network.

These criteria, along with the obvious requirement that none of the 212 existing GRIP-GL gauges could be used as additional testing basins, functioned to eliminate more than 1000 streamflow gauges in the region from consideration and hence resulted in a relatively small sample size of additional test basins.

### 2.2.1 Comparison of different routing structures

This analysis aims to investigate the sensitivity of SR model prediction quality to the chosen delineation method (i.e., the
routing network source and spatial resolution). BasinMaker postprocessing functions were applied to simplify the initial NALRP routing network for the 8 large Non-GRIP-GL basins. Firstly, small lakes were removed by using the same minimum lake area threshold as the GRIP-GL routing product ($5 \text{ km}^2$). Secondly, subbasins were merged by specifying the MDA threshold. For each basin, we delineated 7 routing networks which are defined as follows.

1.  Mimic GRIP-GL routing, by using the same discretization strategy as the GRIP-GL routing product.
2.  *NALRP_10%*, the MDA threshold was calculated as 10 percent of each basin's total drainage area.
3.  *NALRP_100*, the MDA threshold is 100 $\text{km}^2$ for all basins.
4.  *NALRP_300*, the MDA threshold is 300 $\text{km}^2$ for all basins.
5.  *NALRP_500*, the MDA threshold is 500 $\text{km}^2$ for all basins.
6.  *NALRP_800*, the MDA threshold is 800 $\text{km}^2$ for all basins.
7.  *NALRP_1000*, the MDA threshold is 1000 $\text{km}^2$ for all basins.

### 2.3 Performance metrics

In this study, the Kling–Gupta efficiency (KGE, Gupta et al., 2009) is used to evaluate the performance of the lumped LSTM model and the proposed SR model. KGE measures the degree of correspondence between two time series (e.g., observations versus a model prediction of those observations) and here it is computed for daily average streamflow time series. KGE is
defined as follows in Eq. (1):

$$KGE = 1 - \sqrt{(r-1)^2 + (\beta-1)^2 + (\alpha-1)^2}, \qquad (1)$$

where $r$ is the Pearson correlation coefficient which measures the linear correlation between the observed time series and predicted time series, $\beta$ denotes the bias term which indicates whether the model is prone to overestimate or underestimate the streamflow, and $\alpha$ denotes error in flow variability. The range of KGE is $(-\infty, 1]$, where KGE = 1 signifies a perfect
prediction. It is common to use KGE = 0 as the threshold for determining whether the model exhibits good predictive





performance (Knoben et al., 2019). On the other hands, in the GRIP-GL project paper, Mai et al. (2022) carefully argues that in general, a KGE less than 0.48 would be considered a poor model and models with higher KGEs are medium or higher quality.

### 2.4 Experimental design

Three sets of experiments are used to evaluate the quality of the SR model. The first experimental set involved implementing the SR model on the four small Non-GRIP-GL basins. The main objective of this task was to validate the predictive capabilities of the lumped LSTM model in estimating streamflow at a local subbasin-level (LSTM extrapolation to small basins). This was conducted by testing the lumped LSTM on basins that are much smaller than the minimum drainage area in the training dataset. Additionally, we also implemented a single-subbasin routing model in Raven to ensure our approach
to push local streamflow into Raven worked as expected.

In the second set of experiments, we utilized the SR model to predict streamflow on the 212 GRIP-GL basins. This task aims to evaluate the overall performance of the SR model as compared to the lumped LSTM. Following the evaluation scheme of the GRIP-GL project, the KGE was calculated separately for temporal validation (trained locations and untrained period), spatial validation (untrained locations and trained period), and spatiotemporal validation (untrained locations and untrained
period). Moreover, for validation basins (untrained locations), we calculated the KGE for the whole study period (2001-2017). As mentioned earlier, only the GRIP-GL common routing product was used as the routing structure in this task.

For the third set of experiments, the SR model was applied to the 8 large Non-GRIP-GL basins. These basins are equivalent to the validation basins in the GRIP-GL project, where we had no previous knowledge or experience applying the LSTM. Each basin was tested with the 7 routing structures described in Section 2.2.1, in order to investigate the impacts of using
alternative routing networks.

For all experiments, the KGE metric was calculated for the lumped LSTM model predictions and for the semi-distributed SR model predictions, respectively. The lumped LSTM is the benchmark model for comparing with the integrated SR model.

### 3 Results and discussion

Regarding all hydrographs (line plots) in this section, only the data from January 2009 to December 2012 were plotted (i.e.,
2 years in calibration period and 2 years in validation period) for a better visualization, and the displayed KGE for each basin was calculated for the whole study period (2001 - 2017). Note that the lumped LSTM and SR models both make predictions starting from the year of 2001 as their LSTMs takes the year of 2000 as the initial input sequence.

### 3.1 Extrapolating LSTM to small Non-GRIP-GL basins

The lumped LSTM prediction quality on the four small basins (69 - 95 km$^2$) is quite good with KGE values for these new
test locations of 0.785, 0.812, 0.634 and 0.489, respectively. These KGEs compare favourably with the median GRIP-GL



validation performance level reported in Mai et al. (2022) of 0.767 for the same LSTM applied to much larger basins. Figure 3 shows the daily observed streamflow, the predicted streamflow from the lumped LSTM and SR model at each of these four small basins. The predictions from the SR model (blue) are not visible because, as expected, they are the virtually the same as the predictions from the lumped LSTM (red). This can be explained by the fact that the delineation of these small basins

would result in a single subbasin with no spatial discretization, and a single-subbasin Raven routing model would produce the same daily average streamflow as the lumped LSTM inputs to the routing model.

Overall, these results indicate the lumped LSTM adequately extrapolates to much smaller basins than those it was trained for and the translation of LSTM-predicted streamflow into the routing model is correct.

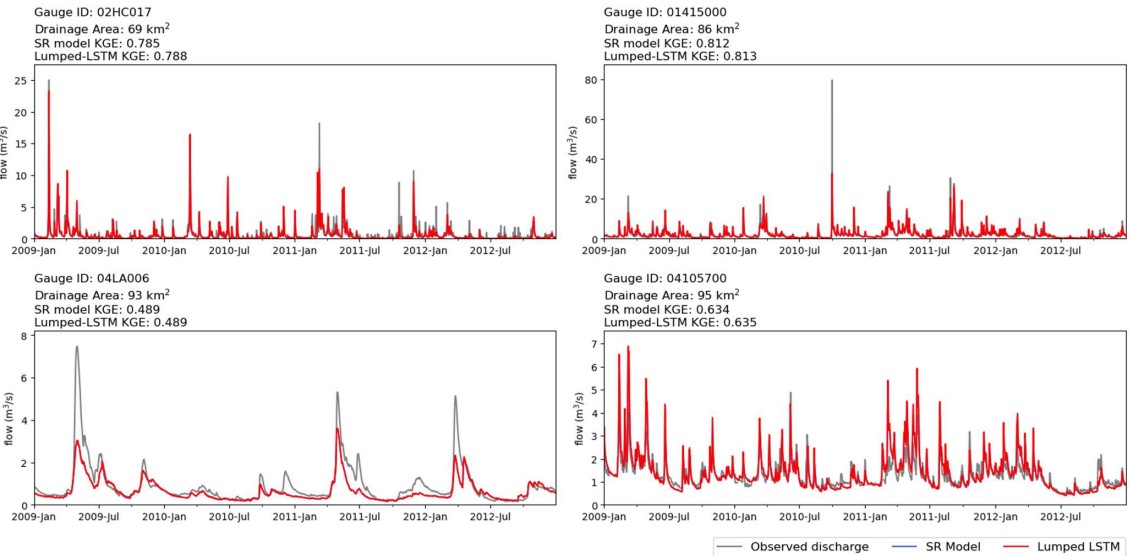


**Figure 3.** Comparison of observation (grey), lumped LSTM prediction (red) and SR model prediction (blue) for the 4 small Non-GRIP-GL basins. KGE values over the period from 2001 to 2017 are given for each basin.

### 3.2 Comparing lumped LSTM and SR model on GRIP-GL basins

The SR modelling approach should be able to outperform the lumped LSTM for basins with large drainage areas, because

spatially distributed modelling will mitigate the information loss caused by feeding the lumped LSTM with basin-averaged features, and the SR model will take advantage of maintaining such spatial heterogeneity. Therefore, the results are evaluated in two ways: results first include all basins (corresponding to the validation experiment in the GRIP-GL study), while the second way focuses solely on basins larger than 1000 km$^2$.

The performance results of the lumped LSTM and the SR model at GRIP-GL basins are summarized in Table 2 and visually

compared in Fig. 4. In the temporal validation experiment, the predictions by the SR model shows a comparable level of





quality to that of the lumped LSTM. The interquartile range of the lumped LSTM is slightly narrower than that of the SR model, suggesting less variability in the KGE score distribution. The results indicate that the lumped LSTM better captures the temporal trends and seasonal patterns at trained locations, while the SR model, relying on an uncalibrated process-based routing model, shows no improvement (slightly reduced KGEs) relative to the lumped LSTM results for both large (>1000

km$^2$) and small (<1000 km$^2$) basins. However, the SR model results for all basins are better than the best of 12 physically-based/inspired GRIP-GL hydrological model (see Table 2).

In the spatial validation and spatiotemporal validation experiments, both the SR model and the lumped LSTM exhibit similar performance degradation relative to temporal validation performance (considering all basins). These two experiments primarily focus on assessing the models' robustness in predicting streamflow in an ungauged basin scenario (where no local

streamflow observations were used to train the model). It is worth mentioning that the LSTM prediction at each subbasin is practically also a prediction of streamflow in an ungauged basin (refer to the first task described in Section 3.1).

As with the temporal validation, when considering all basins in spatial or spatiotemporal validation, the SR model shows no real difference in median KGEs relative to the lumped LSTM results. However, the advantage of the SR model in larger basins becomes apparent in these untrained locations. As compared to the lumped LSTM, for large basins over 1000 km$^2$, the

median KGE of the SR model is 0.072 KGE units higher in spatial validation, and 0.054 KGE units higher in spatiotemporal validation. In contrast for smaller basins, the lumped LSTM remains equal (spatial validation) or slightly better than the SR model (spatiotemporal validation). Furthermore, the SR model results are substantially better than the best of 12 physically-based/inspired GRIP-GL hydrological models for all basins. This is notable given that seven of these 12 models in GRIP-GL were spatially-distributed (not lumped) and utilized the same routing network discretization as the SR model for each basin.

Figure 5 displays the time series of observations, model predictions, and KGE scores, for representative GRIP-GL validation basins. Figure 5A shows the hydrographs of the two basins where the SR model demonstrates the largest improvement (better by more than 0.5 KGE units) compared to the lumped LSTM. Among them, 02KF005 (Ottawa River at Britannia) is the largest basin studied in the GRIP-GL project, with almost a 90000 km$^2$ drainage area. The other basin is approximately 6923 km$^2$ in size (WSC gauge 02LG005, Gatineau Riviere Aux Rapides Ceizur). The deficiency of the lumped LSTM model

is evident as it fails to capture the peaks and seasonal variations at these two large watersheds and the lumped LSTM predicts constantly low flow throughout the study period. Figure 5B shows compares hydrographs at WSC gauges 02HM010 (Salmon River at Tamworth) and 02LB007 (South Nation River at Spencerville), where the prediction accuracy of the SR model shows the worst degradation relative to the lumped model (by 0.17 and 0.12 KGE units). These two gauges are small (588 km$^2$ broken into 9 subbasins in the SR model and 277 km$^2$ broken into 3 subbasins in the SR model) and within the

range of training basin sizes. It is important to note that the spatial resolution of RDRS-v2 dataset is 10 km by 10 km, thereby each grid covers an area of approximately 100 km$^2$ and so breaking these small basins up into a handful of smaller subbasins is likely unnecessary in terms of representing the spatial rainfall patterns. In general, the SR model aligns with the overall pattern of the observed streamflow, but it tends to underestimate peak flows and the lumped LSTM predicts larger peaks on these small basins. This underestimation of peak flow events is not evident in most of the high-quality SR model





hydrographs and in fact in the larger two basins shown in Fig. 5C, the SR model is predicting higher peaks than the lumped
LSTM. Fig. 5C shows the two basins where the SR model achieves the highest KGE scores (both over 0.9 and both large
basins) and these are both notably improved over the lumped LSTM. The hydrographs of two basins where the SR model
achieves the lowest KGE scores (both around 0) are shown in Fig. 5D. It is evident that both lumped LSTM and SR model
were unable to simulate flow in these basins which, according to the observed hydrograph, appear to be substantially
impacted by regulation. The failure of both models in regulated basins is not surprising given none of the LSTM attributes
measure or indicate the degree of regulation within a basin.

**Table 2**. Median KGE for prediction performance of the lumped LSTM and the SR models.

| Validation experiment | Number of basins | Best of other 12 models in GRIP-GL* | Lumped LSTM | SR Model |
|---|---|---|---|---|
| Temporal Validation | 141 | 0.790 | **0.819** | 0.804 |
| Temporal Validation (drainage area **over** 1000 km$^2$) | 58 | 0.807 | **0.839** | 0.830 |
| Temporal Validation (drainage area **below** 1000 km$^2$) | 83 | 0.768 | **0.807** | 0.794 |
| Spatial Validation | 71 | 0.610 | 0.767 | **0.779** |
| Spatial Validation (drainage area **over** 1000 km$^2$) | 25 | 0.589 | 0.708 | **0.780** |
| Spatial Validation (drainage area **below** 1000 km$^2$) | 46 | 0.628 | 0.778 | **0.779** |
| Spatiotemporal Validation | 71 | 0.589 | **0.744** | 0.732 |
| Spatiotemporal Validation (drainage area **over** 1000 km$^2$) | 25 | 0.590 | 0.665 | **0.719** |
| Spatiotemporal Validation (drainage area **below** 1000 km$^2$) | 46 | 0.615 | **0.758** | 0.733 |

*The 12 models are physically-based / physically-inspired models.


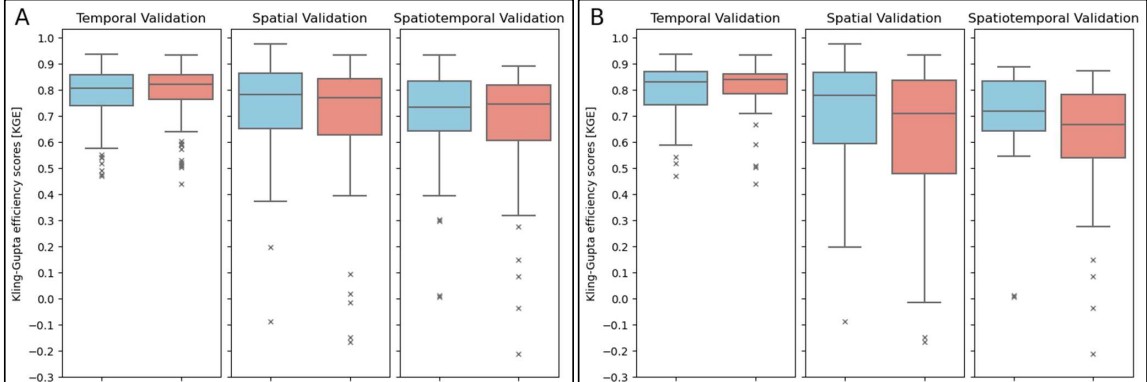

**Figure 4.** Boxplots of KGE validation scores for the SR model (blue) and the lumped LSTM (red). (A) The results of all basins
participating in each validation experiments. (B) The results of basins with a size larger than 1000 km$^2$.




**Figure 5.** Comparison of time series of observation (grey), lumped LSTM prediction (red) and SR model prediction (blue) for
representative GRIP-GL validation basins, over the selected period from 2009 to 2012. KGE values over the whole study period from
2000 to 2017 are given for each basin. (A) The 2 basins that show the most significant improvement as compared to the lumped LSTM.
(B) The 2 basins that show the most significant degradation as compared to the lumped LSTM. (C) The two basins where the SR model
achieves the highest KGE scores. (D) The two basins where the SR model achieves the lowest KGE scores.





### 3.3 New testing basins and impact of routing network delineation

Eight large basins (not used in the GRIP-GL study) were identified as suitable additional independent testing basins according to the criteria in Section 2.2 in order to conduct further comparisons between the lumped LSTM and the SR model built with varying routing networks. Like the GRIP-GL validation basins, neither model was trained on these eight new basins.

Fig. 6 summarizes the comparative results and shows the KGE of each model in each of the eight basins. From Fig. 6, it can

be seen that the choice of the delineation method (routing network resolution) has a minor impact on prediction performance. In general, the differences in overall KGE scores among the 7 different resolution routing networks are not significant at each basin. This could be attributed to the consistent representation of lakes across all the routing networks (all resolutions retain lakes more than 5 km$^2$ in area in the network) combined with the crucial role these lakes play in modelling the transport of water.

In terms of relative model performances, the SR model outperforms the lumped LSTM model in 7 out of the 8 tested basins (improved by an average of 0.160 KGE units, using the GRIP-GL routing network resolution). This result strongly reinforces the findings in Section 3.1, showing that the SR model tends to exhibit better relative performance in basins with larger drainage areas. The SR model achieved the worst KGE score in the basin identified by the USGS gauge 05129115 (Vermilion River near Crane Lake), and this is the only basin where the lumped LSTM (KGE score of 0.716) outperformed

the SR model (KGE scores from 0.534 to 0.600). As depicted in Fig. 6, the KGE scores at this basin exhibit a gradual decrease as the resolution of the routing network becomes coarser, such as with NALRP_800 and NALRP_1000. Figure 7 shows the hydrographs for USGS gauge 05129115 and while it is evident that while the SR model successfully captures the timing of the peaks, it consistently underestimates their magnitude in comparison to the observation and the lumped LSTM.

The degradation in performance could be attributed to the heavy presence of lakes within this basin (as shown in Fig. 8A).

Among the 8 tested basins, 05129115 stands out with the largest fraction of its area covered by lakes, accounting for approximately 14% of its total area. In contrast, the SR model shows significant improvement in basin 02KJ004, which has the second highest proportion of lake coverage, approximately 10% (as depicted in Fig. 8B). However, the lake areas are vastly different in these two basins. Lakes in 05129115 are concentrated mainly as one very large lake in the middle of the basin, whereas the lakes in 02KJ004 are more numerous, smaller, and generally long and narrow. On the other hand, the SR

model obtains a substantial improvement over the lumped LSTM (by over 0.347 KGE units) in a lake-sparse basin gauged by USGS station 04260500 (see Fig. 8C). Notably, this is also the additional testing basin where the SR model achieved the highest KGE score of 0.894.



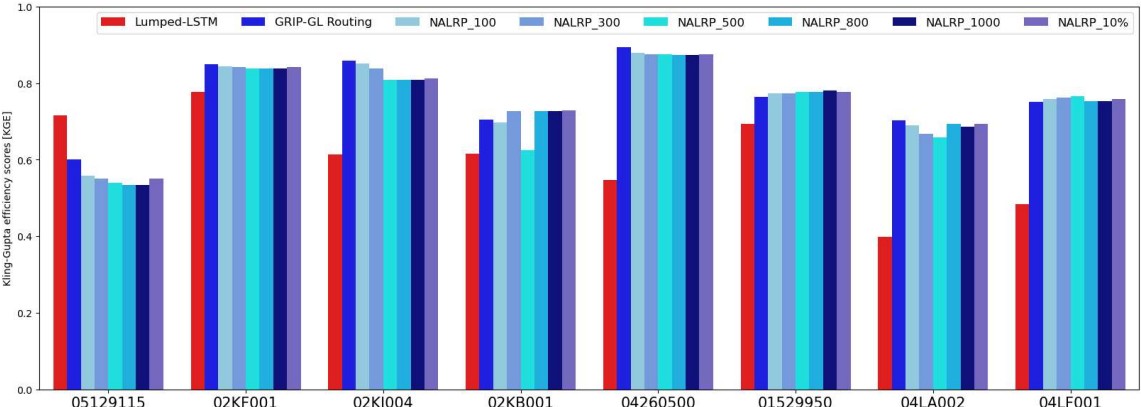

**Figure 6.** Overall model performances as adopting different routing network delineations in the 8 large Non-GRIP-GL basins, during the whole study period 2000-2017. The basins are sorted from left to right in ascending order according to their sizes, from smallest (2243 km$^2$) to largest (5570 km$^2$).

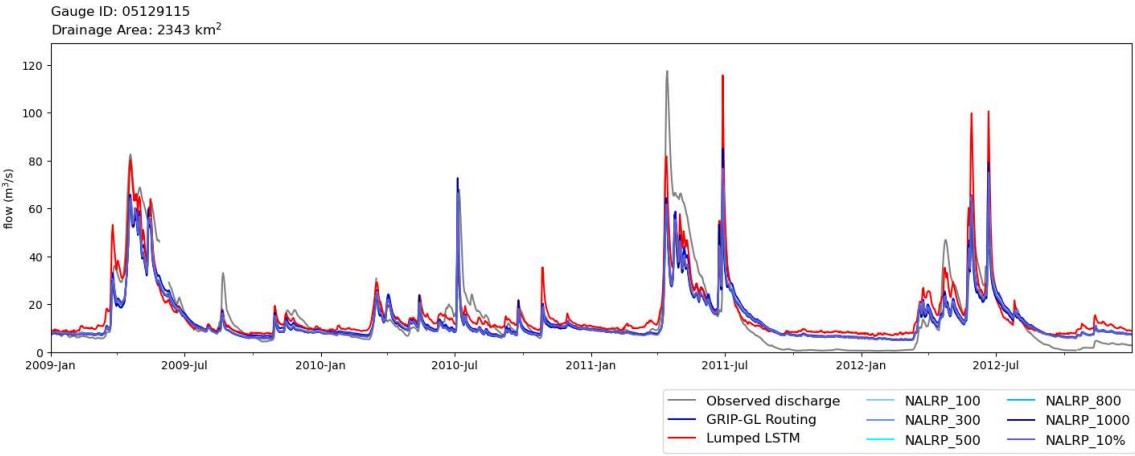

**Figure 7.** Comparison of the time series of observation (grey), lumped LSTM prediction (red) and SR model predictions with different routing networks for the Vermilion River basin 05129115. Note that the lines of NALRP time series all follow extremely similar trends as the GRIP-GL Routing series and thus are not distinguishable.





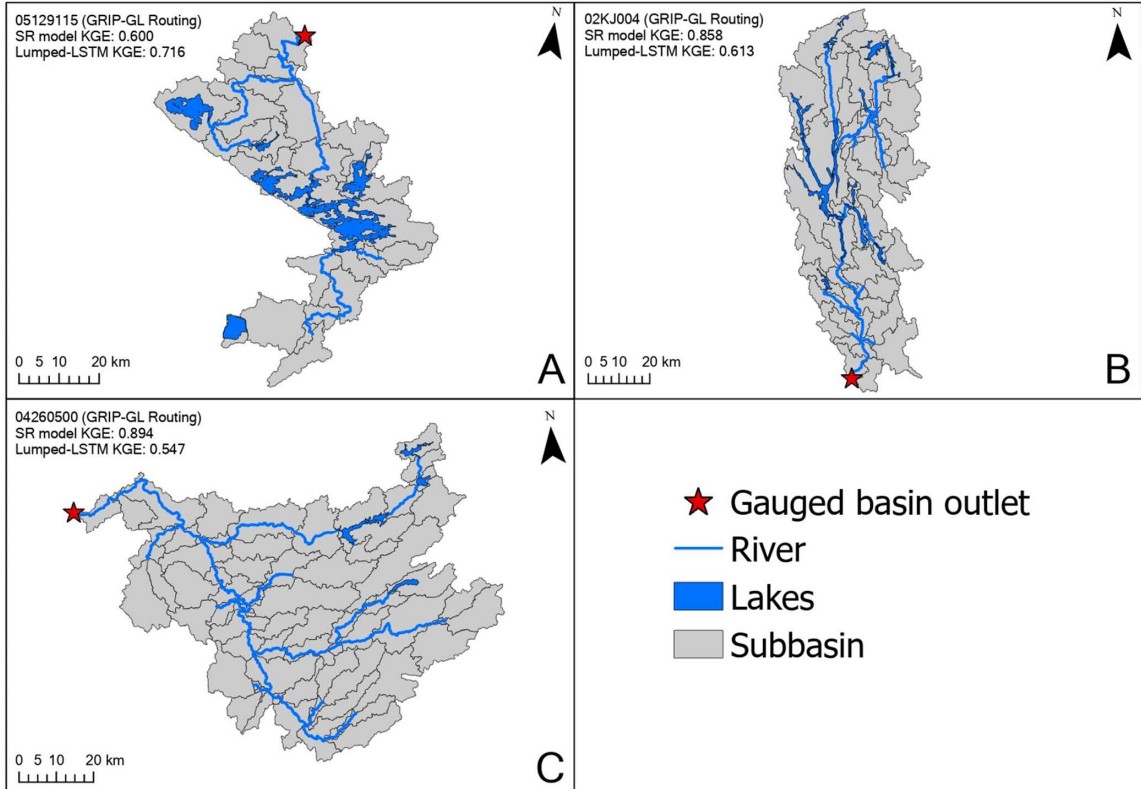

**Figure 8.** Comparison of the routing networks of 3 large Non-GRIP-GL basins. (A) Gauged basin 05129115. (B) Gauged basin 02KJ004 and (C) Gauged basin 04260500. The three illustrated routing networks were delineated to mimic the GRIP-GL routing delineation strategy.

## 4 Conclusions

In this study, we proposed a hybrid modelling approach named the Spatially Recursive (SR) model that aims to enhance the accuracy of streamflow predictions made by lumped data-driven models. For a basin of interest, a regionally-trained lumped LSTM is used to predict the local streamflow at the subbasin-level (as delineated in the basin's lake-river routing network), and then a process-based hydrological routing-only model simulates the transport of local streamflow from the subbasin outlet to the basin outlet. The novelty of the SR model is threefold: (1) It considers the spatial variability of input variables at finer spatial resolution by having smaller response units than the training dataset (i.e., from basin-scale to subbasin-scale); (2) It integrates physically-based hydrological routing with the data-driven learning for enhanced streamflow prediction in large, ungauged basins; (3) It operates without the need for further fine-tuning, parameter transfer, or parameter calibration, given the trained LSTM is available.





Three experiments were conducted to examine the applicability and performance of SR model. First, we validated the concept of predicting streamflow at local subbasin-level with an LSTM trained using much larger basins. This was done by

predicting streamflow at 4 small testing basins (< 100 km²) which were used as mimic local subbasins. The results revealed that the lumped LSTM can indeed be applied for predicting streamflow in basins below the minimum drainage area threshold of the training dataset. Subsequently, the SR model was evaluated on 212 basins from the GRIP-GL project. The results showed that the SR model is comparable to lumped LSTM in terms of overall performance. The SR model exhibits a noticeable advantage in predicting streamflow in large basins (> 1000 km²), which suggests that incorporating spatially

distributed inputs can be beneficial to the hydrological modelling in large basins, due to the fact that the spatial heterogeneity is naturally more significant in larger regions. The improvements in large basins of the SR model were achieved with only a small degradation in relative to lumped LSTMs in the small basins (200 km² to 1000 km²).  In the third task we investigated the impacts of the routing network delineation by testing the SR model in 8 additional large independent testing basins (2243 km² to 5570 km²). Results clearly show the substantial performance gains of the SR model over the lumped LSTM in seven

of the eight basins and these performance gains are not sensitive to a range of routing network configurations. The results further indicate that these performance gains happen in basins with both large and very small fractions of the basin covered by lakes. Importantly, these improvements of the SR model relative to the lumped data-driven model did not require calibration or additional training after the original lumped LSTM was trained. As such, if the hydrological routing model parameters were also trained on the suite of calibration basins like the lumped LSTM, we would expect to see even better SR

model performance.

The findings of this study highlight the importance of considering spatially distributed inputs to streamflow prediction and demonstrate a new way that data-driven models can benefit from such information. This research opens up new avenues for future research regarding hybrid modelling in hydrology, by having a process-based model functioning as the postprocessor of the data-driven model. Future refinement of the purposed SR modelling approach should focus on two key aspects: the

training strategy of the lumped data-driven predictor (e.g., larger dataset, different neural networks etc.), and the calibration of hydrological routing-related parameters.

**Data and code availability**

All code used to implement and validate the models will be available in a dedicated Zenodo repository following the completion of the peer-review process. The GRIP-GL calibration data are made available on the Federated Research Data

Repository (FRDR; https://doi.org/10.20383/103.0598) and the access procedure for GRIP-GL validation data is also described there. The BasinMaker library is available at http://hydrology.uwaterloo.ca/basinmaker/. The Raven hydrologic modelling framework is available at http://raven.uwaterloo.ca.



**Author contributions**

QY implemented the lumped LSTM and SR models, conceptualized the study, and wrote the first draft of the manuscript.
BT conceptualized the study and experimental design and wrote parts of the manuscript. HS implemented the routing-only model in Raven. MH led the iterative application of BasinMaker software to discretize the basins. JM was the lead author on the GRIP-GL study upon which this work depended so heavily, and her contributions ensured the lumped LSTM was implemented correctly. Both JM and JL commented initially on parts of the experimental design and helped polish the manuscript.

**Financial support**

This research has been supported by the NSERC, ECCC and Global Water Futures (GWF) Project.

**Competing interests**

The authors declare that they have no conflict of interest.

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
