# Peer review of "Enhancing LSTM-based streamflow prediction with a spatially distributed approach"

_Hydrology and Earth System Sciences, 2023_

## Referee Comment (RC2)

Thank you for allowing me the opportunity to review your paper. From my understanding, these are the main points of your manuscript:

- You created SR models to created better lumped predictions by connecting LSTM predictions to routing modules.
- There is an option of including/excluding reservoirs.
- The model is trained exclusively on basins within the GL region.
- SR models perform better when performing Spatial validation, while a lumped LSTM does better at the temporal validation.

This manuscript is grammatically well-written and conveys a good message. I recommend the paper is accepted with minor revisions. I have listed Major and Minor comments below, but I believe the major comments are easy to address and do not require a change in the experiment. My biggest issue is with the novelty in the paper (see major comment 4).

Best,

Tadd Bindas

**Major Comments:**

1. There is a main paragraph explaining the importance of large-scale training of LSTMs and how individual catchment LSTMs are not suitable for comparison. However, it seems you're still limiting your data to only regional data from the Great Lakes region. While this is better than training an LSTM on one basin, it is best to give the model *all* of the data you can, including basins outside of the GL region.

   Have you trained your SR, or lumped, LSTM on all CONUS basins, or done any work regarding that?

2. Figure 1 is confusing as the numbering doesn't align with how the graph is read. It reads like it's circular, everything is somehow mapping to itself. I suggest redoing this figure as it would be easier for the reader to understand the SR workflow.

3. What loss function was used? While it makes sense to have a lot of your information in a supplemental paper, adding this information would be helpful to readers as they would know what information is used in training your lumped LSTM.

4. I'm unsure of the novelty here. After reading the paper, and the novelty statement, multiple times. I'm not convinced that what you're doing hasn't already been done. It appears you're only routing streamflow from a model created in Mai et al. (2022).

   I hate to share/recommend reading my own research as it can come off as scummy, but I have an accepted manuscript that has a very similar workflow what you're doing here (Training a LSTM that was created based on another paper's work, applying the LSTM to smaller catchments, routing those outputs, and comparing the routed flows against the

lumped predictions. My paper's novelty is that our Muskingum-Cunge routing is a differentiable model). The reason I'm sharing my work is because you may not be aware that I had used a very similar set up in my research. The paper has been under review for ~15 months and only available in preprint. See below for the Title and DOI.

*Improving large-basin river routing using a differentiable Muskingum-Cunge model and physics-informed machine learning*
10.22541/essoar.168500246.67971832/v1

To be fair to your work, I do think that your paper presents an angle that has not been done yet, but is not expressed in the novelty statement:
*"The distinctive aspect of our research lies in the combination of components to form the SR model. The following subsections explain the specifics of the components we selected in this study to demonstrate the spatially recursive modelling approach."*
I believe your work's novelty is both the scale (as my accepted work is done at the catchment scale, *not at the regional scale),* and my work does not have a reservoir module included in the routing. This will ensure your paper correctly identifies what is an expansion of domain knowledge.

**Minor Comments:**

1. Lines 107-108: This sentence reads a little weird. I suggest changing to: "Han et al. (2020) and Mizukami et al. (2016) include examples of hydrologic routing models."
2. Can you make the font bigger for figure 5 axis?

---

## Author Comment (AC1)

**RESPONSE TO RC1 COMMENTS:**

Thanks for the time and effort you invested in reviewing the manuscript, we appreciate the detailed comments you provided. In response to the five questions you raised (*repeated here in red italic text*), please find our detailed responses below in regular black text:

*Overall, this is a very interesting study considering how to integrate deep learning models with process-based models. The paper introduces a Spatially Recursive (SR) model based on GRIP-GL data. The model first trains a basin LSTM and then uses the trained LSTM to simulate the flow of subbasins. Finally, it obtains the ultimate basin flow through a Routing-only mode. While the aspects mentioned in the paper are not individually new approach, this combination demonstrates a certain level of innovation.*

We are pleased you recognize the innovation here.

*RC1, C1: [The "Routing-only mode" is an important part of the document, and perhaps a flowchart would help readers better understand its workflow. Additionally, the document mentions that the input for the Routing-only mode is hourly data. The question is how the authors transform data from the LSTM into hourly data.]*

The workflow of the routing mode was briefly explained in Section 2.1.3. We don't see how a flow chart can enhance the workflow understanding given that for "Routing-only mode", the workflow is really only input file conversion and then a Raven model configured only for routing. Instead, in response to your suggestion, precise details of the Raven routing-only mode configuration, such as routing method algorithms, will be added to the appendix in the revised manuscript ensuring readers can precisely replicate our Raven Routing-only configuration.

The daily LSTM-predicted outputs are in units of millimeters per day (mm/d). We transformed the daily LSTM predictions to hourly routing simulation inputs by assuming the precipitation intensity is constant over the day. That is, for a given date, the LSTM-predicted streamflow is assigned to all 24 hourly time steps. We will ensure this transformation is noted in the revised Section 2.1.3.

*RC1, C2: [Including a map of the study area would indeed be beneficial for readers unfamiliar with the GRIP-GL project.]*

Thanks for your valuable suggestion, we will add a detailed map of the study region in the revision, featuring the classification of calibration and validation basins. The draft new map is shown below:

[Figure]

*RC1, C3: [The paper mentions that initially, basin data is used to train the LSTM, which is then applied to predict streamflow in subbasins using the subbasins' input data. Since there is sufficient subbasin data available, the question is why not directly train a subbasin-specific LSTM for predicting subbasin streamflow, which could then enable the prediction of basin streamflow through Routing-only mode?]*

We assume the reviewer is asking us why we did not train specifically on subbasin-scale (i.e., very small local drainage basins) watersheds. That would be ideal since we assume that finer resolution (i.e., smaller-scale data) would result in higher quality of predictions at that scale. The primary reason we did not do that is because we wanted to show that our SR model built with an existing LSTM regional streamflow model, trained with watersheds beyond the subbasin scale delineated in the routing product (The GRIP-GL calibration basins are ranging from 200 km$^2$ to 16000 km$^2$, while the average subbasin scale is approximately 131 km$^2$ in the GRIP-GL routing product) can easily be augmented with an uncalibrated hydrological routing approach to enhance streamflow prediction in larger watersheds where the lumped LSTM (with no hydrological routing) predictions are not as good. In this way, our SR model as implemented is a lower bound estimate of the optimal performance our SR modelling approach could achieve. For example, if we instead trained our LSTM by adding additional subbasin scale watersheds (< 200 km$^2$) to the training set, we can assume performance would improve, or at the very least, not degrade. Overall, our work shows hydrological modellers a reliable way to improve LSTM-based streamflow predictions without having to do any more training or calibration.

One of the practical reasons we did not train with subbasin-scale watersheds is that the baseline lumped LSTM we duplicated from our previous work in Mai et al. (2022), specifically targeted gauged watersheds

greater than 200 km$^2$. Given the findings of Kratzert et al. (2023) showing that training on more basins is always better, we would caution against the assumption that developing any subbasin-scale specific LSTM (targeting and training on only watersheds smaller than 200 km$^2$ for example) would be optimal within our SR modelling approach. Another practical reason we did not do this is that the discharge observations are unavailable for most of the subbasins delineated in the GRIP-GL routing product and the North American Lake-River Routing product. Even though we have sufficient training features (i.e., dynamic forcings and static attributes) for the subbasins, we do not have the target variable (i.e., subbasin streamflow).

We will integrate some of the above rationale into our revised manuscript to highlight this question of utilizing subbasin-scale watersheds in the subbasin level LSTM development.

*RC1, C4: [In the discussion in section 3.1, for smaller basins, the spatial segmentation might still represent a sub-basin. In this scenario, is the structure of the Spatially Recursive (SR) model still the same as it would be for multiple sub-basins, or is the Routing-only mode not used under these conditions?]*

Yes, for smaller basins, the delineated routing network would only contain one subbasin which is geometrically identical to the basin outline. The structure of the SR model is still the same as it would be for multiple sub-basins, that is, the routing simulation will still be applied on these smaller basins. In such case, the routing model only functions to take the subbasin streamflow (LSTM-predicted) as an input and it would be directly flushed without delay to the basin outlet, making it equivalent to a lumped prediction. In other words, routing model application does not change LSTM-predicted inputs for such a case.

We will ensure our revision in section 3.1 makes the above answer clear.

*RC1, C5: [What is the role of Figure 8? Providing different delineations of the routing network within a single basin might better help in understanding the impact of routing network delineation.]*

As explained in the paragraph from line 400 – 417, Figure 8 is to demonstrate how our findings might relate to different lake densities in the different watersheds. On the other hand, Figure 2 is used to demonstrate the impact of different delineations of the routing network in a single basin. In response to this comment, we will revise the caption of Figure 8 to make its purpose clear to the readers.
* * *
We look forward to hearing your thoughts on the revised manuscript and hope for a positive outcome. Should you require any further information or clarification, please do not hesitate to contact us.

Thank you once again for your time and expertise.

Qiutong and co-authors

**References**

Kratzert, F., Gauch, M., Klotz, D., and Nearing G.: Never train an LSTM on a single basin, EartharXiv [preprint], https://doi.org/10.31223/X57090, 2023.

Mai, J., Shen, H., Tolson, B. A., Gaborit, É., Arsenault, R., Craig, J. R., Fortin, V., Fry, L. M., Gauch, M., Klotz, D., Kratzert, F., O'Brien, N., Princz, D. G., Rasiya Koya, S., Roy, T., Seglenieks, F., Shrestha, N. K., Temgoua, A. G. T., Vionnet, V., and Waddell, J. W.: The Great Lakes Runoff Intercomparison Project Phase 4: The Great Lakes (GRIP-GL), Hydrol Earth Syst Sci, 26, 3537–3572, https://doi.org/10.5194/hess-26-3537-2022, 2022.

---

## Author Comment (AC2)

**RESPONSE TO RC2 COMMENTS:**

Thanks for the time and effort you invested in reviewing the manuscript, we appreciate the detailed comments you provided. In response to the six questions you raised (*repeated here in red italic text*), please find our detailed responses below in regular black text:

*RC2, C1: [There is a main paragraph explaining the importance of large-scale training of LSTMs and how individual catchment LSTMs are not suitable for comparison. However, it seems you're still limiting your data to only regional data from the Great Lakes region. While this is better than training an LSTM on one basin, it is best to give the model all the data you can, including basins outside of the GL region. Have you trained your SR, or lumped, LSTM on all CONUS basins, or done any work regarding that?]*

No, we have not done that. We agree with you given the findings of Kratzert et al. (2023) showing that training on more basins is always better. The primary reason we did not train a new model on CONUS basins in this study is because we wanted to show that our SR model built with an **existing** LSTM regional streamflow model (i.e., the GRIP-GL lumped LSTM), can easily be augmented with an uncalibrated hydrological routing approach to enhance streamflow prediction in larger watersheds where the lumped LSTM (with no hydrological routing) predictions are not as good. In this way, no need for researchers to train a new lumped LSTM on new dataset.

*RC2, C2: [Figure 1 is confusing as the numbering doesn't align with how the graph is read. It reads like it's circular, everything is somehow mapping to itself. I suggest redoing this figure as it would be easier for the reader to understand the SR workflow.]*

We will rearrange the figure in the revised manuscript so the numbering appears more natural with how one would read the graph and we will augment the caption to be more descriptive and help guide readers through this. We note that the arrows in the workflow are definitely not circular. However, data and components are used multiple times and hence multiple arrows.

*RC2, C3: [What loss function was used? While it makes sense to have a lot of your information in a supplemental paper, adding this information would be helpful to readers as they would know what information is used in training your lumped LSTM.]*

The loss function is the same as the loss function introduced in Kratzert et al. (2019) (see the equation in the screenshot below), which is effectively the averaged Nash–Sutcliffe efficiency (NSE) value across the calibration/training basins.

$$\text{NSE}^* = \frac{1}{B} \sum_{b=1}^{B} \sum_{n=1}^{N} \frac{(\widehat{y}_n - y_n)^2}{(s(b) + \epsilon)^2}, \qquad (13)$$

where $B$ is the number of basins, $N$ is the number of samples (days) per basin $B$, $\widehat{y}_n$ is the prediction of sample $n$ ($1 \leq n \leq N$), $y_n$ is the observation, and $s(b)$ is the standard deviation of the discharge in basin $b$ ($1 \leq b \leq B$), calculated from the training period. In general, an entity-aware deep

In response to your suggestion, and details of the LSTM configuration, such as the hyperparameters and training variables, will be added to the appendix in the revised manuscript.

*RC2, C4: [I'm unsure of the novelty here. After reading the paper, and the novelty statement, multiple times. I'm not convinced that what you're doing hasn't already been done. It appears you're only routing streamflow from a model created in Mai et al. (2022). I hate to share/recommend reading my own research as it can come off as scummy, but I have an accepted manuscript that has a very similar workflow what you're doing here (Training a LSTM that was created based on another paper's work, applying the LSTM to smaller catchments, routing those outputs, and comparing the routed flows against the lumped predictions. My paper's novelty is that our Muskingum-Cunge routing is a differentiable model). The reason I'm sharing my work is because you may not be aware that I had used a very similar set up in my research. The paper has been under review for ~15 months and only available in preprint. See below for the Title and DOI.*

*Improving large-basin river routing using a differentiable Muskingum-Cunge model and physics-informed machine learning 10.22541/essoar.168500246.67971832/v1*

*To be fair to your work, I do think that your paper presents an angle that has not been done yet but is not expressed in the novelty statement: "The distinctive aspect of our research lies in the combination of components to form the SR model. The following subsections explain the specifics of the components we selected in this study to demonstrate the spatially recursive modelling approach."*

*I believe your work's novelty is both the scale (as my accepted work is done at the catchment scale, not at the regional scale), and my work does not have a reservoir module included in the routing. This will ensure your paper correctly identifies what is an expansion of domain knowledge.]*

Below we will detail the four key differences (two of which are noted by the reviewer above) between our SR model approach and the approach described in the Bindas et al. (2023) unpublished preprint.

Our SR model applies an LSTM to directly generate routing model inputs at the spatial scale of the routing model. Specifically, our LSTM directly predicts local subbasin streamflow appearing at the subbasin outlet and these are directly the inputs to our routing model. We have demonstrated that the SR model is robust and works without user intervention when the routing model scale, and hence the scale at which the LSTM predicts local subbasin streamflow, is varied.

In contrast, the approach in Bindas et al (2023) is fundamentally different as their LSTM streamflow predictions do not match the spatial scale of their routing model and hence require an intermediate, and somewhat unclear, scale-specific parameterization to translate in inputs for their 2 km reach length routing model. Specifically, they generate HUC10 subbasin level streamflow predictions with their LSTM and then rescale each of these within the subbasin to lateral inflows of each 2 km reach. How to adjust their rescaling approach for either a new shorter or longer reach and/or a larger or smaller subbasin scale (not HUC10 subbasins) is not reported on.

Beyond the above difference, the Bindas et al. (2023) preprint only demonstrates they can enhance lumped LSTM predictions of larger watersheds (>2000 km$^2$) *in a single catchment* when their routing model is trained in that catchment. Our method is *at the regional*, not catchment. scale, and demonstrates our untrained SR approach (uncalibrated routing model) enhances LSTM predictions in 27 out of 44 larger watersheds (>2000 km$^2$) spread across the GRIP-GL study region.

There is also a third key difference. Given an existing applicable lumped LSTM model for streamflow, the Bindas et al. (2023) preprint uses a physics-informed machine learning approach to improve upon the

lumped LSTM, but that improvement requires additional ML-model training. Our approach requires no such complex retraining, out of reach to hydrological modellers not trained in machine learning and is thus simpler and directly applicable for all hydrologic modellers. It is not clear the approach in Bindas et al. (2023) would work as successfully at the regional scale, with more spatially variable routing conditions. Particularly, given the scale-specific rescaling approach noted above which may only work in their 5260 km$^2$ catchment case study. Clearly, if we calibrated our routing model to each of our 200+ catchments, our SR model results would improve even further. Even without calibrating our routing model in each catchment like Bindas et al. (2023) did, we note the magnitude of our streamflow prediction improvements in larger watersheds (>2000 km$^2$, see our Figure 6 in our manuscript showing KGE increases of roughly 0.05 to 0.3 KGE units in 7 of 8 basins) seem to be equal or larger than the Bindas et al. (2023) improvements which are reported as an NSE metric increase to 0.857 from 0.801. We acknowledge the different units here but believe our point stands.

Another less important novel aspect of our manuscript is that we utilize a routing model that explicitly includes lakes, unlike the BIndas et al. (2023) preprint whose routing model does not simulate lakes explicitly.

We will add a streamlined summary of these 4 differences into our revised manuscript more clearly indicating the relative novelty of our work.

*RC2, C5: [Lines 107-108: This sentence reads a little weird. I suggest changing to: "Han et al. (2020) and Mizukami et al. (2016) include examples of hydrologic routing models."]*

Thanks for the suggestion, we will rephrase that sentence in the revised manuscript.

*RC2, C6: [Can you make the font bigger for figure 5 axis?]*

Thanks for the suggestion, we will remake the figure in the revised manuscript.

We look forward to hearing your thoughts on the revised manuscript and hope for a positive outcome. Should you require any further information or clarification, please do not hesitate to contact us.

Thank you once again for your time and expertise.

Qiutong and co-authors

**References**

Kratzert, F., Klotz, D., Shalev, G., Klambauer, G., Hochreiter, S., and Nearing, G.: Towards learning universal, regional, and local hydrological behaviors via machine learning applied to large-sample datasets, Hydrol Earth Syst Sci, 23, 5089–5110, https://doi.org/10.5194/hess-23-5089-2019, 2019.

Kratzert, F., Gauch, M., Klotz, D., and Nearing G.: Never train an LSTM on a single basin, EartharXiv [preprint], https://doi.org/10.31223/X57090, 2023.

---

## Author Response (AR1)

**Point-by-point response to the reviews of our manuscript "Enhancing LSTM-based streamflow prediction with a spatially distributed approach" including a list of all relevant changes made in the manuscript. Please note that our published initial responses to the review comments on the HESS website are not included here. We believe our finalized responses below, which often repeat much of the initial responses published, stand alone.**

**RESPONSE TO EDITOR COMMENTS:**

*Thank you for your responses to the two sets of reviews. We were expecting a third review, but the reviewer was unable to submit it on time, therefore we will proceed with these two reviews and your replies. I appreciate your detailed responses, and especially the clarification of the novelty and significance of the manuscript. Based on these replies, I would like to invite you to submit a fully-revised manuscript, which will be returned to the reviewers.*

We appreciate the review and believe our revisions described below completely address all the review comments. We are pleased to hear the novelty was clarified for you.

**RESPONSE TO RC1 COMMENTS:**

*RC1, C1: [The "Routing-only mode" is an important part of the document, and perhaps a flowchart would help readers better understand its workflow. Additionally, the document mentions that the input for the Routing-only mode is hourly data. The question is how the authors transform data from the LSTM into hourly data.]*

The workflow of the routing mode is briefly explained in Section 2.1.3. Rather than a flow chart we have added clarity to readers with a supplemental document describing how the routing model works in complete detail. Section S2 of Supplement has enough detail for precisely replicating our routing model. In addition, the minor changes to Figure 1 (and caption) add clarity as well. Finally, we added a sentence to the revised manuscript on line 249 - 251 in Section 2.1.3 answering the above specific question as follows:

> **"We transformed the daily LSTM streamflow predictions to hourly routing simulation inputs by assuming the streamflow is constant over the day. That is, for a given date, the LSTM-predicted streamflow is assigned to all 24 hourly time steps."**

*RC1, C2: [Including a map of the study area would indeed be beneficial for readers unfamiliar with the GRIP-GL project.]*

We have added a detailed map of the study area to the supplemental file of this paper, please see Section S1 of the Supplement.

*RC1, C3: [The paper mentions that initially, basin data is used to train the LSTM, which is then applied to predict streamflow in subbasins using the subbasins' input data. Since there is sufficient subbasin data available, the question is why not directly train a subbasin-specific LSTM for predicting subbasin streamflow, which could then enable the prediction of basin streamflow through Routing-only mode?]*

We assume the reviewer is asking us why we did not train specifically on subbasin-scale (i.e., very small local drainage basins) watersheds. That would be ideal since we assume that finer resolution (i.e., smaller-scale data) would result in higher quality of predictions at that scale. The primary reason we did not do that is because we wanted to show that our SR model built with an existing LSTM regional streamflow model, trained with watersheds beyond the subbasin scale delineated in the routing product (The GRIP-GL calibration basins are ranging from 200 km$^2$ to 16000 km$^2$, while the average subbasin scale is approximately 131 km$^2$ in the GRIP-GL routing product) can easily be augmented with an uncalibrated hydrological routing approach to enhance streamflow prediction in larger watersheds where the lumped LSTM (with no hydrological routing) predictions are not as good. In this way, our SR model as implemented is a lower bound estimate of the optimal performance our SR modelling approach could achieve. For example, if we instead trained our LSTM by adding additional subbasin scale watersheds (< 200 km$^2$) to the training set, we can assume performance would improve, or at the very least, not degrade. Overall, our work shows hydrological modellers a reliable way to improve LSTM-based streamflow predictions without having to do any more training or calibration.

We synthesized our extended answer above and have added the following sentence to the opening paragraph of our LSTM section (line 149 – 152 in the revised manuscript):

> **"The decision to employ an existing lumped LSTM for streamflow prediction, rather than attempting to add more basins and retrain a new LSTM, was intentional as we wanted to demonstrate explicitly that the proposed spatially distributed methodology works to improve upon an existing lumped LSTM without the need for LSTM retraining."**

And in the conclusion paragraph (line 509 - 511) describing the limitations of this study we added as follows:

> **"The regional LSTM could be purpose-built to train on a sufficient number of small basins better matching the subbasin-level spatial scale at which the lumped LSTM would be applied within the SR model (e.g., 131 km$^2$ as the average subbasin size of the GRIP-GL routing product)."**

We would also note for the reviewer that given the findings of Kratzert et al. (2023) showing that training on more basins is always better, we would caution against a potential assumption that developing any subbasin-scale specific LSTM (targeting and training on only watersheds smaller than 200 km$^2$ for example) would be optimal within our SR modelling approach. Another practical reason we did not do this is that the discharge observations are unavailable for most of the subbasins delineated in the GRIP-GL routing product and the North American Lake-River Routing product. Even though we have sufficient training features (i.e., dynamic forcings and static attributes) for the subbasins, we do not have the target variable (i.e., subbasin streamflow).

*RC1, C4: [In the discussion in section 3.1, for smaller basins, the spatial segmentation might still represent a sub-basin. In this scenario, is the structure of the Spatially Recursive (SR) model still the same as it would be for multiple sub-basins, or is the Routing-only mode not used under these conditions?]*

Workflow-wise, the routing-only model is used under these conditions: a gauged basin with a delineated routing network would only contain one subbasin which is geometrically identical to the basin outline. Hence, the routing simulation will still be applied to such a single subbasin routing network and in such a case, the routing model only functions to take the subbasin streamflow (LSTM-predicted) as an input

(constant rate for each 24hrs and input to the model at an hourly timestep) and it would be moved instantly without delay to the basin outlet, making it equivalent to a lumped LSTM basin streamflow prediction at the daily time scale. We should note that none of the GRIP-GL routing networks (Section 3.2) are delineated with a single subbasin.

Section 3.1 (line 351 - 355) is now revised to clarify the above as follows:

> **"The predictions from the SR model (blue) are not visible because, as expected, they are the virtually the same as the predictions from the lumped LSTM (red). This can be explained by the fact that the delineation is geometrically identical to the basin outline (i.e., no spatial discretization and each basin is only a single subbasin). In the routing simulation, the LSTM-predicted subbasin streamflow would be directly flushed without delay to the basin outlet, making it equivalent to a lumped prediction."**

*RC1, C5: [What is the role of Figure 8? Providing different delineations of the routing network within a single basin might better help in understanding the impact of routing network delineation.]*

Figure 8 is to demonstrate how our findings might relate to different lake densities in the different watersheds. On the other hand, Figure 2 is used to demonstrate the impact of different delineations of the routing network in a single basin. In response to this comment, we revised the caption of Figure 8 (line 470 - 473) to make its purpose clear to the readers, as follows:

> **"Figure 8. Comparison of the routing networks of 3 large Non-GRIP-GL basins to illustrate the variation in different lake densities. (A) Gauged basin 05129115 (5 lakes, 14% of basin is covered by lakes). (B) Gauged basin 02KJ004 (11 lakes, 10% of basin is covered by lakes) and (C) Gauged basin 04260500 (4 lake, 4% of basin is covered by lakes). The three illustrated routing networks were delineated to mimic the GRIP-GL routing delineation strategy."**

**Responses to the second reviewer (RC2) starts on next page.**

**RESPONSE TO RC2 COMMENTS:**

*RC2, C1: [There is a main paragraph explaining the importance of large-scale training of LSTMs and how individual catchment LSTMs are not suitable for comparison. However, it seems you're still limiting your data to only regional data from the Great Lakes region. While this is better than training an LSTM on one basin, it is best to give the model all the data you can, including basins outside of the GL region. Have you trained your SR, or lumped, LSTM on all CONUS basins, or done any work regarding that?]*

No, we have not done that. We agree with you and so does Kratzert et al. (2023) who show that training on more basins is always better. The primary reason we did not train a new LSTM model on CONUS basins in this study is because we wanted to show that our SR model built with an existing LSTM regional streamflow model (i.e., the GRIP-GL lumped LSTM), can easily be augmented with an uncalibrated hydrological routing approach to enhance streamflow prediction in larger watersheds where the lumped LSTM (with no hydrological routing) predictions are not as good. In this way, no need for researchers to train a new lumped LSTM on new dataset.

We synthesized the above rationale and included the following sentence in the opening paragraph of our LSTM section (2.1.1) of the paper (line 149 – 152 in the revised manuscript):

> **"The decision to employ an existing lumped LSTM for streamflow prediction, rather than attempting to add more basins and retrain a new LSTM, was intentional as we wanted to demonstrate explicitly that the proposed spatially distributed methodology works to improve upon an existing lumped LSTM without the need for LSTM retraining."**

*RC2, C2: [Figure 1 is confusing as the numbering doesn't align with how the graph is read. It reads like it's circular, everything is somehow mapping to itself. I suggest redoing this figure as it would be easier for the reader to understand the SR workflow.]*

We modified the workflow figure with some rearranging, along with the caption. We would like to note that the arrows in the workflow are (and were not) circular. However, data and components are used multiple times by various components and hence multiple arrows. Please see the revised Figure 1 and caption in the revised manuscript.

*RC2, C3: [What loss function was used? While it makes sense to have a lot of your information in a supplemental paper, adding this information would be helpful to readers as they would know what information is used in training your lumped LSTM.]*

The loss function is the same as the loss function introduced in Kratzert et al. (2019), which is effectively the averaged Nash–Sutcliffe efficiency (NSE) value across the training basins. We made sure to add a supplemental document for this manuscript and in it, we list all information about the lumped LSTM including the loss function equation.

Rather than duplicate the Supplement contents here, we refer you to the Section S3 of the Supplement we submitted along with the manuscript revision to see the complete LSTM documentation.

*RC2, C4: [I'm unsure of the novelty here. After reading the paper, and the novelty statement, multiple times. I'm not convinced that what you're doing hasn't already been done. It appears you're only routing streamflow from a model created in Mai et al. (2022). I hate to share/recommend reading my own research as it can come off as scummy, but I have an accepted manuscript that has a very similar workflow what you're doing here (Training a LSTM that was created based on another paper's work, applying the LSTM to smaller catchments, routing those outputs, and comparing the routed flows against the lumped predictions. My paper's novelty is that our Muskingum-Cunge routing is a differentiable model). The reason I'm sharing my work is because you may not be aware that I had used a very similar set up in my research. The paper has been under review for ~15 months and only available in preprint. See below for the Title and DOI.*

*Improving large-basin river routing using a differentiable Muskingum-Cunge model and physics-informed machine learning 10.22541/essoar.168500246.67971832/v1*

*To be fair to your work, I do think that your paper presents an angle that has not been done yet but is not expressed in the novelty statement: "The distinctive aspect of our research lies in the combination of components to form the SR model. The following subsections explain the specifics of the components we selected in this study to demonstrate the spatially recursive modelling approach."*

*I believe your work's novelty is both the scale (as my accepted work is done at the catchment scale, not at the regional scale), and my work does not have a reservoir module included in the routing. This will ensure your paper correctly identifies what is an expansion of domain knowledge.]*

Thank you for sharing your preprint. Given it was a preprint when we conducted our research and crafted the original manuscript, we were not aware it existed and so everything we have presented was developed without knowing anything of the research in Bindas et al. (2024).

We strongly disagree with your concern about the novelty of our work but certainly understand why you might think that from our original introduction versus the content in your paper. Yes, at the highest level our aim is the same as your paper: enhance prediction accuracy on large basins by combining LSTMs applied at the small spatial scale with physically-based routing. However, how we have approached the problem and our demonstration of the efficacy of our method are completely different than what is done in Bindas et al. (2024). For example, we consider lakes in our routing model, we demonstrate methodological success at the regional level rather than individual watershed level, we do not require the watershed-specific training/calibration of the routing model. Overall, our method is technically simpler to implement.

We will carefully detail these fundamental differences below and then synthesize these more concisely for inclusion in the revised introduction in order to clearly indicate the relative novelty. We think it would be awkward to include all the details in the revised manuscript, but we want to include them in this response file to be sure that you and the Editors are convinced we have produced novel work.

Our SR model applies an LSTM to directly generate routing model inputs *at the spatial scale of the routing model*. Specifically, our LSTM directly predicts local subbasin streamflow appearing at the subbasin outlet and these are directly the inputs to our routing model. We have demonstrated that the SR model is robust and works without user intervention when the routing model scale, and hence the scale at which the LSTM predicts local subbasin streamflow, is varied. Furthermore, we even demonstrated our method works 'out of the box' for routing networks built from different DEMs (see Section 2.1.2 and a restatement

of this fact in a single sentence now added to Section 2.1.4). All of this is achieved without any routing model calibration or any LSTM/ML training beyond the regional pretraining of the employed LSTM.

In contrast, the approach in Bindas et al. (2024) is fundamentally different as their LSTM streamflow predictions do not match the spatial scale of their routing model and hence require an intermediate, scale-specific, and watershed-specific parameterization to translate LSTM streamflow predictions into lateral flow inputs for each of their 2 km reaches in the routing model. Specifically, they generate HUC10 subbasin level streamflow predictions with their pre-existing LSTM and then rescale each of these within the subbasin to lateral inflows of each 2 km reach (based on the derived incremental drainage areas). More significantly, a simple alternative to the inverse-routing approach was required to revert LSTM-predicted runoff at the HUC10 outlets to the time before it enters the river network in each of their 2 km reaches, due to the scale mismatch. Their approach required additional watershed-specific and scale-specific testing and tuning of their inverse routing parameter. Their overall approach to addressing the LSTM scale and routing scale mismatch is not generalizable to new watersheds or different spatial scales and thus requires site-specific fine-tuning followed by watershed-specific training of their differentiable physics-Informed machine learning routing model.

Beyond the above difference, Bindas et al. (2024) report that the differentiable routing model performed noticeably better than the uniform (i.e. lumped) LSTM model at 1 of 3 large (~2000 km$^2$ or greater) internal gauging stations not used in training within *a single 5212 km$^2$ watershed*. This is achieved when their routing model is trained on the watershed outlet and when the training/testing is restricted to a single year in the period of record. The differentiable routing model was also not trained or applied in testing mode for the multi-year period of record available in their watershed. Their body of evidence empirically showing their differentiable routing model is better than the uniform LSTM is limited. On the contrary, our method is *at the regional*, not watershed scale, and demonstrates our *untrained* SR approach (uncalibrated routing model) beats the lumped LSTM in 1) spatial validation over an 10-yr period in 10 of 15 large GRIP-GL basins (KGE improvement in the 10 basins is average of 0.24 KGE units) and 2) spatial validation over a 17-yr period for 7 of 8 additional large non-GRIP-GL basins (KGE improvement in the 7 basins is average of 0.16 KGE units). Our body of empirical evidence is much stronger.

There is also a third key difference. Given an existing applicable lumped LSTM model for streamflow, Bindas et al. (2024) requires additional neural network training using watershed-specific attributes to derive two routing parameters per 2 km reach. On the contrary, our approach does not require such further training nor routing parameter calibration. It is not clear the approach in Bindas et al. (2024) would work as successfully at the regional-scale or larger, where it would encounter much more spatially variable routing conditions. Particularly, the scale-specific rescaling approach noted above was only validated in the 5212 km$^2$ catchment case study.

Another novel aspect of our manuscript is that we utilize a routing model that explicitly includes lakes, while in BIndas et al. (2024) the routing model is based on a river network which does not include lakes explicitly.

Given the above detailed evidence on why we think our contribution is novel, we think the following new paragraph in the introduction section (line 95 - 105) is a fair synthesis helping to highlight this by noting there is more research to do beyond Bindas et al. (2024):

> **"A recent study by Bindas et al. (2024) presents a novel differentiable river routing method to improve the streamflow prediction in a single large basin. They employed a regionally-trained LSTM to predict discharge at the subbasin-level and then map the LSTM predictions to a river network for routing. The results of the final routing model show promise by simulating**

**predicted subbasin-level discharge from a lumped LSTM. However, their study presents limited empirical evidence demonstrating the superiority of their routing model over the lumped LSTM. Specifically, the comparison was conducted within one single basin for a short testing period of one year, and the routing model outperforms the lumped LSTM in one of the three untrained gauges with larger than 2000km$^2$ of drainage area. As for methodology, an intermediate process (scale-specific and basin-specific) is required to translate the LSTM predictions into lateral flow inputs for each reach in the river network. Furthermore, the routing model requires training of a multilayer perceptron network to update the parameters. As such, their basin-scale approach is challenging to directly generalize to new basins, or to different spatial scales for modelling."**

In addition, we have edited our original introduction paragraph about the novelty of our work (line 106 – 113, which appears right after the above new paragraph) as follows:

**"Our study aims to identify an easy-to-implement, generalizable, regional-scale (or larger) approach for applying spatially-distributed inputs to effectively improve upon lumped data-driven streamflow prediction, especially in large, ungauged basins. In pursuit of this goal, we propose the Spatially Recursive (SR) model. The SR model first employs a lumped data-driven prediction model (regionally-trained on a large sample of basins) to predict local streamflow at subbasins discretized from the basin of interest. Then, it utilizes a semi-distributed hydrologic routing-only model, capable of explicit lake simulation, to route subbasin streamflow to the basin outlet. The data-driven prediction model is considered spatially recursive because it is trained at the basin scale and further applied at the subbasin scale to incorporate finer-resolution forcing data and subbasin attributes."**

In the following sections of the revised manuscript, we note in various places what is unique about our method.

Line 132 - 137: We edited the paragraph describing the model coupling in the methodology section:

**"The distinctive aspect of our research lies in the smooth coupling of the LSTM and the routing model. The regionally-trained LSTM is applied to directly produce the local streamflow at the subbasin-scale (i.e., the spatial scale of the routing model), thus no need for scale transformation when we use the LSTM prediction as the direct input to the routing model. Additionally, the SR model does not require any further training/calibration of either of the regional LSTM or the routing model, making it generalizable to any designated gauge or basin outlet within the LSTM training region."**

Line 479 - 483: We edited the novelty statement in the conclusion section to make sense:

**"The novelty of the SR model is threefold: (1) It considers the spatial variability of input variables at finer spatial resolution by having smaller response units than the training dataset (i.e., in our case, this is from basin-scale to subbasin-scale); (2) It integrates physically-based lake-river hydrological routing with the data-driven learning to form a generalizable modelling approach for enhanced streamflow prediction in large ungauged basins; (3) It operates without the need for further fine-tuning, parameter transfer, or training/calibration, given the trained LSTM is available."**

Line 493 - 498: We edited the paragraphs in the conclusion section to better summarize the relative performance of our model, highlighting the empirical evidence:

"The empirical performance improvement over the lumped LSTM is most significant in a PUB context. For the 15 large GRIP-GL validation basins, the median KGE levels over the 10-year training period and the 7-year testing period, are 0.11 and 0.08 KGE units higher, respectively, than those of the lumped LSTM. Importantly, for smaller basins (< 2000 km²), the performance gains for large basins do not result in significant performance drops, as the median KGE difference of SR model and the lumped LSTM were within 0.03 KGE units in all three validation modes."

*RC2, C5: [Lines 107-108: This sentence reads a little weird. I suggest changing to: "Han et al. (2020) and Mizukami et al. (2016) include examples of hydrologic routing models."]*

Thanks for the suggestion, we rephrased the sentence (line 131 – 132 of the revised manuscript), as follows:

"Furthermore, Han et al. (2020) and Mizukami et al. (2016) include examples of vector-based hydrologic routing models."

*RC2, C6: [Can you make the font bigger for figure 5 axis?]*

Done for x-axis and y-axis. Please see Figure 5 in the revised manuscript.

**ADDITIONAL MANUSCRIPT CHANGE BY AUTHORS (NOT IN RESPONSE TO SPECIFIC REVIEW COMMENTS):**

1. In addition to the changes listed above, we clarified the definition of the term 'large basin' in the revised manuscript and made corresponding adjustments. In the initial manuscript, the definition of 'large basin' was inconsistent. We used 1000 km² as the threshold in Section 3.2 for model performance comparison, while employing 2000 km² in Sections 2.2 and Section 3.3 for selecting new testing basins. In an effort to objectively pick a standard value, we reviewed various large-sample hydrology literature more closely and found 2000 km² as a commonly used threshold. Therefore, in the revision, the threshold is consistently set to be 2000 km² throughout the paper (including in Table 2), and we provide the following justification for this in the introduction section (line 82 - 94) as follows:

"Conceptually, the predictive accuracy of lumped hydrological modelling will eventually degrade as basin size increases. This is due to the fact that meteorological forcing inputs are simply not well approximated by assuming they are constant over space. For example, in their paper presenting the CAMELS dataset of attributes for 671 basins in the contiguous United States, Addor et al. (2017) caution against using the largest of these basins for lumped hydrological modelling. They argued that the significance of basin-averaged input attributes diminishes with an increase in basin drainage area, because larger basin tends to necessitate a heightened consideration of spatial heterogeneity, requiring the incorporation of a spatially distributed representation. Nevertheless, the threshold at which basin area leads to poor

**lumped model performance is not precisely known and will likely vary by watershed location. In a study of benchmarking multiple hydrologic models, Newman et al. (2017) excluded basins in the CAMELS dataset over 2000 km2 in drainage area. Additionally, the lumped LSTM modelling study by Kratzert et al. (2019) and the original set of basins in the CARAVAN lumped global large-sample hydrology dataset (Kratzert et al., 2023) both employ 2000 km2 as an upper threshold. Given the past use of the 2000 km2 threshold, we apply this criterion to define a 'large basin' in our study."**

While this change has improved the SR model performance metrics for 'large basins' in Table 2 relative to the initial submission, the improvements do not change any of the conclusions in the manuscript. The SR model works well on aggregate in basins over 1000 km$^2$ (i.e., the initial threshold in first version of the manuscript) and performance for large basins defined by the new threshold of 2000 km$^2$ is somewhat better than the metrics under the previous threshold.

2. We would like to make a point we failed to make in the first version of the manuscript. We strongly believe our SR model performance is a lower bound of that it could be. Our conclusion section has a revised paragraph justifying this as follows (line 505 - 511):

**"Moreover, these improvements of the SR model relative to the lumped data-driven model did not require calibration or additional training after the original lumped LSTM was trained. In fact, the reported results of the SR model reflect a conservative estimate (i.e., lower bound) of performance, considering the following factors: (1) The regional LSTM within the SR model could be improved with additional training data (Kratzert et al., 2024); (2) The SR model routing parameters could be calibrated and regionalized to further improve validation results; and (3) The regional LSTM could be purpose-built to train on a sufficient number of small basins better matching the subbasin-level spatial scale at which the lumped LSTM would be applied within the SR model (e.g., 131 km$^2$ as the average subbasin size of the GRIP-GL routing product). "**

**References**

Bindas, T., Tsai, W. P., Liu, J., Rahmani, F., Feng, D., Bian, Y., Lawson, K., and Shen, C.: Improving River Routing Using a Differentiable Muskingum-Cunge Model and Physics-Informed Machine Learning, Water Resour Res, 60, https://doi.org/10.1029/2023WR035337, 2024.

Kratzert, F., Klotz, D., Shalev, G., Klambauer, G., Hochreiter, S., and Nearing, G.: Towards learning universal, regional, and local hydrological behaviors via machine learning applied to large sample datasets, Hydrol Earth Syst Sci, 23, 5089–5110, https://doi.org/10.5194/hess-23-5089-2019, 2019.

Kratzert, F., Gauch, M., Klotz, D., and Nearing G.: Never train an LSTM on a single basin, EartharXiv [preprint], https://doi.org/10.31223/X57090, 2023.